# Cost-effectiveness of the SMILE intervention compared with usual care for people with severe mental illness: A randomized clinical trial

**Mohamed El Alili**[1,2]*, **Berno van Meijel**[3,4,5], **Maurits W. van Tulder**[6], **Marcel Adriaanse**[1]

1 Department of Health Sciences, Vrije Universiteit Amsterdam, Faculty of Science, Amsterdam Public Health Research Institute, Amsterdam, The Netherlands, 2 Zorginstituut Nederland, Diemen, The Netherlands, 3 Inholland University of Applied Sciences, Department of Health, Sports & Welfare, Research Group Mental Health Nursing, Amsterdam, The Netherlands, 4 Amsterdam UMC (VUmc), Department of Psychiatry, Amsterdam Public Health research institute, Amsterdam, The Netherlands, 5 Parnassia Psychiatric Institute, Parnassia Academy, The Hague, The Netherlands, 6 Department Clinical, Neuro- & Developmental Psychology, Faculty of Behavioural and Movement Sciences, Vrije Universiteit Amsterdam, Amsterdam, The Netherlands

* m.elalili@vu.nl

**Data Availability Statement:** All relevant data are within the manuscript and its Supporting information files.

## Abstract

### Objectives

Only studying effectiveness of lifestyle interventions for people with severe mental illness (SMI) is insufficient for policy making. As budgets for healthcare are limited, policy makers face the problem of allocating scarce healthcare resources. Cost-effectiveness studies are needed, but currently cost-effectiveness studies of lifestyle interventions for people with SMI delivered in ambulatory care are limited. The aim of this current study was to evaluate the cost-effectiveness of a lifestyle intervention for people with SMI living in the Dutch community in comparison with usual care.

### Methods and findings

An economic evaluation was performed using a societal perspective alongside the Severe Mental Illness Lifestyle Evaluation (SMILE) pragmatic cluster randomized controlled trial. The SMILE lifestyle intervention is a one-year, group-based intervention delivered by trained mental healthcare workers. Costs, body weight change and quality of life were assessed at baseline, 6 and 12 months. Mixed models were used to estimate incremental costs and effects between the treatment group and the usual care group. Overall, the SMILE intervention resulted in lower total costs compared to the usual care group (-€719, 95% CI -7133; 3897). The effect difference between the intervention and usual care groups was -3.76 (95% CI -6.30; -1.23) kilograms for body weight and -0.037 (95% CI -0.083; 0.010) for QALYs.

### Conclusions

Overall, the SMILE intervention resulted in lower total costs compared to the usual care group and was cost-effective for body weight change. However, the SMILE intervention

**Funding:** This study was funded by grant 80–84300–98-72012 from the Netherlands Organization for Health Research and Development (ZonMW: https://www.zonmw.nl/nl). The funder had no role in the design and conduct of the study; collection, management, analysis, and interpretation of the data; preparation, review, or approval of the manuscript; and decision to submit the manuscript for publication. MCA was the main applicant, BJGvM and MWvT were co-applicants. ME had no role in no role in securing funding.

**Competing interests:** The authors have declared that no competing interests exist.

does not seem cost-effective with regards to QALYs. More cost-effectiveness studies in other countries and other settings are needed to gain further insight into the cost-effectiveness of lifestyle interventions for people with SMI.

## Introduction

People with severe mental illness (SMI) have a lower life expectancy of 15 to 20 years compared to the general population [1–3]. Though the number of unnatural deaths due to suicide and accidents in patients with schizophrenia or bipolar disorder is high [4, 5], physical conditions account for approximately 70% of deaths [4, 6]. Approximately 33% of patients with bipolar disorder and 25% of patients with schizophrenia die from cardiovascular disease [7–9]. Risk factors that contribute to cardiovascular disease such as unhealthy nutrition patterns, low physical activity, smoking, genetic vulnerability, and low access to healthcare are common within this patient population [1, 10–13]. On top of that, antipsychotic drugs, which are frequently part of treatment, are associated with an increased risk of gaining weight, dyslipidaemia and diabetes mellitus [12]. However, people with SMI receive fewer interventions and benefit less from treatment of physical diseases than the general population [1, 5].

In The Netherlands, outpatient mental health care for people with SMI living in the community is commonly offered by Flexible Assertive Community Treatment (FACT) teams. Routine care within FACT-teams consists of multidisciplinary treatment and care by individual case management in the community. Care is determined by the needs of the patient; unstable patients with a high risk of relapse receive intensive assertive outreach care [14]. Awareness of the importance of integrating physical health care within routine mental health care for this population is rising. An increasing number of clinical guidelines for mental health care for people with SMI include a focus on physical health [13]. Although many studies have been conducted [13, 15], evidence for long-term sustainability of lifestyle interventions for people with SMI is still limited [16–19], and implementation into routine care is lacking [19, 20]. Also, little is known regarding the cost-effectiveness of lifestyle interventions within this population as studies often do not describe cost data or address cost-effectiveness [13, 17, 21, 22]. Studying the effectiveness of lifestyle interventions alone is insufficient for policy making. As budgets for healthcare are limited, policy makers face the problem of allocating scarce healthcare resources. Adding lifestyle interventions to routine care may increase costs, such as costs of training staff and extra hours spend on intervention sessions. It is important to know whether these additional costs are worth the potential added benefits. By performing economic evaluations of lifestyle interventions, we gain insight in the cost-effectiveness of these interventions and assist policy makers in establishing priorities within the cost-constrained healthcare budgets. Hence, the aim of this current study was to evaluate the cost-effectiveness of a lifestyle intervention for people with SMI living in the Dutch community in comparison with usual care.

## Methods

### Study design

An economic evaluation from a societal perspective was performed alongside a pragmatic cluster randomized controlled trial (the SMILE-study). Follow-up duration was one year. The design of the SMILE-study was described elsewhere [23], as well as the process evaluation [24],

experiences of people with SMI and health care professionals with the SMILE intervention [25] and the effectiveness of the SMILE intervention [26]. The protocol was approved by the medical ethics committee of the VU University Medical Centre (NL60315.029.17, registration number 2017.418), registered in the Netherlands Trial Register (Identifier: NTR6837, https://trialsearch.who.int/Trial2.aspx?TrialID=NTR6837). The data as registered in the Netherlands Trial Register are automatically included in the International Clinical Trial Registry Platform of the World Health Organization (WHO). The authors confirm that all ongoing and related trials for this intervention are registered. Measurements took place at baseline, six and 12 months. Weight was also measured at three months. Questionnaires were completed on hard copy to make the questionnaires more accessible for participants. All participants received 10 euros for each of the measurements performed at baseline, six and 12 months.

## Setting and participants

The study was conducted within 21 FACT-teams from eight mental healthcare centres in the Netherlands. The recruitment for the trial was conducted from the 1st of January 2018 to the 1st of January 2019 with a follow-up of one year. Participants were adults (18 years or older), had a BMI of 27 or higher, and received care from one of the participating FACT-teams. All participants provided written consent. Exclusion of participants was based on cognitive impairments that could interfere with actively participating in the intervention, contra-indications such as an acute psychiatric crisis (assessed by the FACT-teams' healthcare professionals), inability to communicate in the Dutch language, or (planning for) pregnancy or breastfeeding.

## Randomization

We performed cluster randomization at the level of FACT-teams. One centre participated with only an intervention team, as their control team dropped out of the study after randomisation. A statistician without further involvement in the study performed the randomization using a computerized random number function in Microsoft Excel.

## Sample size

The pragmatic cluster randomized controlled trial was powered to detect a mean difference of 4 kg in weight reduction after 1 year between the intervention and usual care group [27, 28]. Using a power of 0.80 and an alpha of 0.05, two groups of 100 participants were needed to detect a difference of 4 kg. For this cluster randomized trial, an ICC of 0.01 was assumed. In addition, taking into account a drop-out rate of 20% [27–29], the goal was to recruit 260 participants in total.

## Intervention

The SMILE intervention is a group-based intervention which focusses on lifestyle changes with an emphasis on a healthy diet and physical activity. In total, there were 30 group sessions throughout the year. People with SMI followed group sessions weekly during the first six months and monthly during the last six months. Group sessions were given by trained FACT-team healthcare professionals. The intervention was primarily modelled after the STRIDE intervention [28, 30, 31]. A more detailed description of the SMILE intervention can be found elsewhere [23–26, 30].

## Economic evaluation

Both a cost-effectiveness analysis (CEA) and cost-utility analysis (CUA) were performed with body weight change (in kilograms) and quality adjusted life years (QALYs) as primary outcomes, respectively. For QALYs, the EuroQol-5D-5L (EQ-5D-5L) questionnaire was used [32]. We converted utility scores using the Dutch EQ-5D tariff with a range from –0.446 to 1, where negative utilities indicate a health state that is valued as worse than death, and 1 indicates full health [33]. The EQ-5D-5L contains five dimensions (mobility, self-care, usual activities, pain/discomfort and anxiety/depression) with five response levels (no problems, slight problems, moderate problems, severe problems or extreme problems) [34].

## Costs

Costs were measured from a societal perspective. Costs were converted to 2020, if necessary, using Dutch consumer price indexes [35]. The time horizon of the economic evaluation was 1 year, which means that discounting is not necessary.

**Intervention costs.** We calculated the costs of the intervention using a bottom-up approach. Intervention costs consisted of all costs that were associated with the execution of the intervention. This included the intervention materials, costs of group leaders per hour multiplied by the hours spend on their preparation per group session and the time of the group sessions themselves, time spent on training for group leaders, and travel costs to each group session.

**Healthcare utilization costs and productivity costs.** To reduce the burden of participating in the study, we used an adapted and shorter questionnaire (based on the TIC-P questionnaire) for measuring healthcare utilization and productivity costs [36]. In this questionnaire, we collected data regarding quantity of healthcare that participants received, such as contacts with FACT-team staff (i.e., nurses and psychologists), physiotherapists, general practitioners, outpatient visits, and informal care from relatives or friends. Dutch standard cost prices were used when available [37]. For prices that were not available at least five sources were consulted (e.g., job sites) and an average was calculated. For productivity loss, costs of sick leave were calculated using the mean wage per hour per gender and 5-years age group. Presenteeism costs were calculated by multiplying hours with the mean wage per hour per gender and age group.

In addition, medication use was collected based on the most recent pharmacy overview lists of participants. All medication was grouped into respective medication groups based on the Anatomical Therapeutic Chemical (ATC) code [https://www.whocc.no/]. To calculate medication costs per group we used the highest Dutch costs of each medication based on prices reported by the Dutch National Health Care Institute (Zorginstituut Nederland; www.medicijnkosten.nl). We calculated average medication costs by multiplying the daily dose and the average costs of the respective medication group.

## Statistical analysis

The cost-effectiveness analyses were conducted according to the intention-to-treat principle. Missing data were imputed using Multiple Imputation with Chained Equations (MICE) [38]. Cost and effect data were assumed to be missing at random, which means that missing observations are explained by observed variables [39]. The imputation model included outcome variables and predictor variables that either differed at baseline, were related to missing data, or were associated with the outcome (see Table 1 for variables included in the imputation model). To account for the skewed distribution of cost data, predictive mean matching was used in MICE [40]. The number of imputed datasets was increased until the loss of efficiency was less than 5%, resulting in five imputed datasets [40]. Each of the imputed datasets was analysed

**Table 1. Multiply imputed costs per patient and overall effects over time.**

| | | Intervention group (N = 126) | Usual care group (N = 97) | Main analysis (N = 223) | Including outlier* (N = 224) |
|---|---|---|---|---|---|
| *Outcomes* | | Mean (SE) | | Mean difference (95%CI)§ | Mean difference (95%CI)§ |
| Body weight (kg) | T0 (baseline) | 101.28 (1.75) | 102.39 (1.83) | -3.76 (-6.77; -0.75) #, † | -3.66 (-6.66; -0.66)#, † |
| | T1 (3 months) | 99.22 (1.87) | 102.12 (1.91) | | |
| | T2 (6 months) | 97.97 (1.89) | 102.17 (1.82) | | |
| | T3 (12 months) | 97.66 (2.09) | 102.52 (2.04) | | |
| QALY | | 0.64 (0.020) | 0.73 (0.025) | -0.037 (-0.085; 0.012) † | -0.033 (-0.081; 0.014) † |
| *Healthcare costs* | | | | | |
| Primary care | | 255 (59) | 138 (18) | 117 (41; 216) | 118 (41; 216) |
| Secondary care | | 507 (60) | 245 (43) | 262 (122; 399) | 264 (124; 403) |
| Health contacts | | 406 (71) | 486 (143) | -80 (-413; 177) | -102 (-443; 154) |
| Health contacts within FACT team | | 1881 (150) | 1927 (221) | -46 (-575; 417) | -44 (-578; 409) |
| Admission | | 1340 (602) | 2436 (1100) | -1096 (-3772; 844) | -1071 (-3761; 849) |
| Home care | | 1026 (207) | 2750 (2006) | -1724 (-6519; 504) | -3601 (-10276; 212) |
| Informal care | | 1076 (219) | 465 (142) | 611 (165; 1117) | 616 (164; 1114) |
| Medication | | 1630 (182) | 1794 (190) | -164 (-499; 158) | -157 (-495; 161) |
| Intervention | | 930 (-) | 0 (-) | 930 (-) | 930 (-) |
| *Lost productivity costs* | | | | | |
| Absenteeism FCM | | 1173 (363) | 1190 (417) | -17 (-1083; 986) | -4 (-1085; 976) |
| Absenteeism HCA | | 1425 (474) | 1408 (536) | 17 (-1330; 1305) | 31 (-1309; 1326) |
| Presenteeism | | 545 (312) | 23 (18) | 522 (155; 1182) | 522 (149; 1178) |
| Loss of unpaid work | | 100 (20) | 106 (32) | -6 (-69; 50) | -5 (-67; 49) |
| **Total societal costs FCM** | | 10870 (1117) | 11559 (2583) | -690 (-6348; 3386) | -2534 (-9724; 2486) |
| **Total societal costs HCA** | | 11122 (1159) | 11778 (2619) | -656 (-6350; 3561) | -2498 (-9550; 2662) |

*One case was added in the control group, bringing the total sample size to n = 224. This concerned the outlier, which was due to a patient in this group that received 20 hours of home care per day for a consecutive 24 weeks.

§Uncertainty around cost differences estimated using the non-parametric bootstrap with 5000 replications (bias-corrected intervals). The presented cost differences are unadjusted.

#Overall effect over time (12 months).

†Covariates included in the mixed model for weight were treatment indicator, baseline weight, gender, primary diagnosis of mental disorder, having a partner and smoking. The mixed model for QALYs was adjusted for baseline utility.

Multiple imputation model consisted of variables that differed at baseline, were related to missing data or were associated with the outcome: gender, diagnosis of mental disorder, smoking at baseline, smoking at 12 months, baseline costs of health contacts outside FACT team, age, weight at baseline, body mass index at baseline, education, home care costs at baseline, utility at baseline, heart rhythm disorder and diabetes. The imputation procedure was stratified for treatment arm and the cluster indicator variable was added to the imputation model to adjust for clustering in the imputation procedure.

FACT = flexible assertive community treatment, FCM = friction cost method, HCA = human capital approach, QALY = Quality adjusted life-year, SE = standard error, 95%CI = 95% confidence interval.

separately as described below. Results from the multiple datasets were pooled using Rubin's rules [41].

Mixed models were used to estimate incremental costs and effects between the treatment group and usual care group. This means that we corrected for clustering in the data by allowing the intercepts to vary between clusters (i.e., random intercepts model). For costs and QALYs, a two-level structure was used where FACT-teams and patients represented the first and second level, respectively. For the difference in body weight, an additional level accounted for repeated observations within patients (i.e., patients' body weight measures at different time

points) [42, 43]. Hence, a three-level structure was used where these repeated patients' observations were nested within patients, which were nested within FACT-teams. This allowed for estimation of an overall effect over time [44]. QALYs were adjusted for baseline utilities [45]. Weight was additionally adjusted for confounders (see Table 1 for list of confounders). Incremental Cost-Effectiveness Ratios (ICERs) were calculated by dividing the incremental costs by incremental effects. Bias-corrected bootstrapping was used to estimate statistical uncertainty (5000 replications). Statistical uncertainty surrounding ICERs was illustrated by plotting the bootstrapped cost-effect pairs on a cost-effectiveness plane (CE-plane). Cost-effectiveness acceptability curves (CEACs) were estimated, which demonstrate the probability that the intervention is cost-effective compared to usual care for a range of different ceiling ratios (i.e., the willingness-to-pay threshold for one point effect extra) [46]. CEACs were estimated using the parametric p-value approach for incremental net-monetary benefits (INMBs) [47]. In the Netherlands, the generally used willingness-to-pay threshold for healthcare ranges between 20,000 and 80,000 Euros per QALY gained depending on the burden of disease [48]. For outcome measures such as weight, no formal willingness-to-pay threshold has been determined. Analyses were performed in StataSE 16® (StataCorp LP, CollegeStation, TX, US).

To check the robustness of the results, two sensitivity analyses were performed. The economic evaluation was performed without adjustment for confounders (SA1) and an analysis was performed including an outlier (SA2). The outlier concerned a patient that received 140 hours of home care in 24 weeks, which had a large impact on the estimation of home care costs.

## Results

### Patient flow and sample characteristics

The study population included 11 lifestyle intervention FACT-teams (126 participants) and 10 usual care FACT-teams (98 participants). In the main analysis of the economic evaluation, one outlier was removed from the usual care group. This was due to a patient that received 20 hours of home care per day for a consecutive 24 weeks. Further details about patient flow and baseline characteristics are reported elsewhere (discrepancies in the number of patients analysed between the economic evaluation and the effectiveness study are due to the use of multiple imputation) [26]. See Fig 1.

### Costs

For most cost categories, costs were lower in the intervention group than in the usual care group, but not statically significant (see Table 1). When costs were higher in the intervention group compared to the usual care group, these were significantly higher, except for absenteeism costs as estimated by the human capital approach. The largest difference in costs between the intervention and the usual care group was found for home care (i.e., -€1724), while the smallest difference in costs between intervention and usual care groups was found in loss of unpaid work (i.e., -€6). Total societal costs were €690 lower (95% CI -6348; 3386) in the intervention group compared to the usual care group; this difference was not statistically significant.

### Cost-effectiveness

The difference in *body weight change* (in kilograms) between the intervention group and the usual care group was -3.76 kilograms, indicating that over time the intervention resulted in a weight loss of -3.76 kilograms (95% CI -6.30; -1.23) more than usual care (see Table 2). This

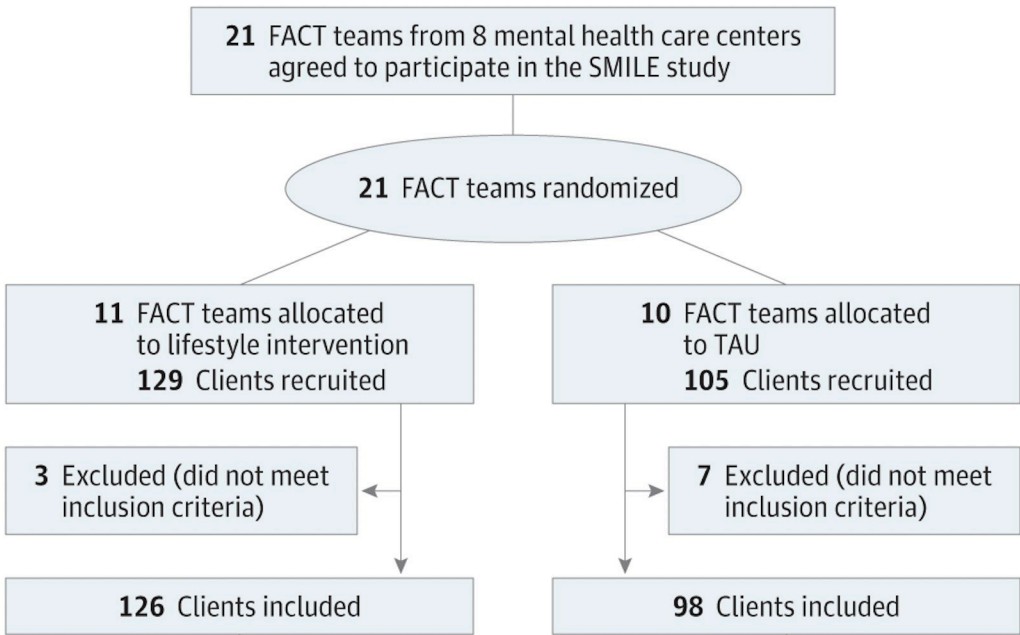

**Fig 1. CONSORT participant flow diagram of recruitment for the SMILE study.**

difference was statistically significant. For body weight, the intervention was dominant compared to the control group, since effects were larger and costs were lower. The CE-plane (Fig 2a) shows that most of the bootstrapped cost-effect pairs are situated in the south-east quadrant of the plane confirming more weight loss (i.e., weight loss is a favourable outcome, hence a loss of weight is inverted on the CE-plane) and lower costs in the intervention group as compared to the usual care group. The CEA curve (Fig 2b) shows that the probability that the intervention is cost-effective in comparison with usual care is 0.59, 0.92 and 1.00 at willingness-to-pay values of €0, €1000 and €10,000 per kilogram body weight loss, respectively.

**Table 2. Results of the cost-effectiveness analyses.**

| Outcome | Mean cost difference (95% CI)* | Mean effect difference (95% CI) | ICER | Distribution of the cost-effectiveness plane (%) | | | |
|---|---|---|---|---|---|---|---|
| | | | | NE | SE | SW | NW |
| **Main analysis** | | | | | | | |
| Body weight | -719 (-7133; 3897)* | -3.76 (-6.30; -1.23) *,† | 191 | 39% | 61% | 0% | 0% |
| QALYs | -719 (-7133; 3897)* | -0.037 (-0.083; 0.010) *,† | 19481 | 1% | 4% | 57% | 38% |
| **Unadjusted analysis (SA1)** | | | | | | | |
| Body weight | -719 (-7133; 3897)* | -4.16 (-8.17; -0.16)* | 173 | 38% | 60% | 1% | 1% |
| QALYs | -719 (-7133; 3897)* | -0.081 (-0.15; -0.017)* | 8867 | 0% | 0% | 61% | 39% |
| **Including outlier (SA2)** | | | | | | | |
| Body weight | -2993 (-10852; 2800)* | -3.66 (-6.18; -1.14)*, † | 818 | 18% | 82% | 0% | 0% |
| QALYs | -2993 (-10852; 2800)* | -0.033 (-0.079; 0.012) *,† | 89376 | 1% | 6% | 76% | 17% |

CI = confidence interval,, ICER = incremental cost-effectiveness ratio, NE = north-east quadrant, NW = north-west quadrant, SA1 = sensitivity analysis 1,

SA2 = sensitivity analysis 2, SE = south-east quadrant, SW = south-west quadrant, QALY = quality adjusted life-year

*Difference in costs and effects is adjusted for clustering of FACT team. The difference in weight is additionally adjusted for repeated observations.

†Covariates included in the mixed model for weight were treatment indicator, baseline weight, gender, diagnosis of mental disorder, having a partner and smoking. The mixed model for QALYs was adjusted for baseline utility.

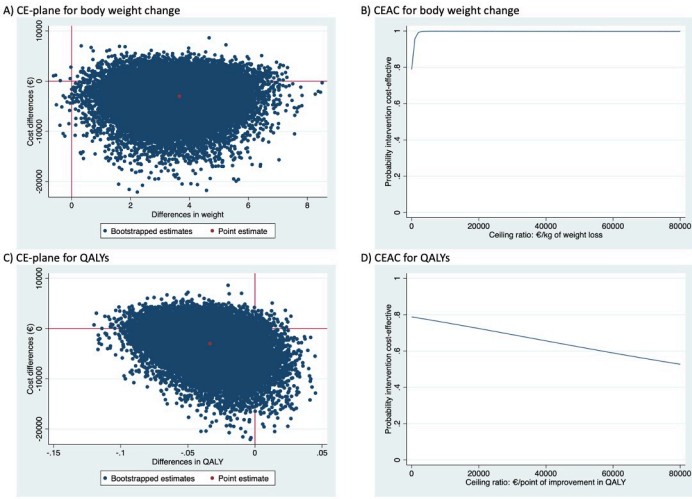

**Fig 2. Cost-effectiveness planes and cost-effectiveness acceptability curves.**

The difference in *QALYs* between the intervention and usual care group was -0.037 (95% CI -0.083; 0.010), which was not statistically significant. The ICER for QALYs was 19481, indicating that €19481 is saved in the intervention group in comparison with the usual care group when 1 QALY is lost. The CE-plane (Fig 2c) shows that the bootstrapped cost-effect pairs are mainly located in the south-west quadrant confirming the smaller effects and lower costs in the intervention group as compared to the usual care group. The CEA curve (Fig 2d) shows that the probability that the intervention is cost-effective in comparison with usual care is 0.59, 0.50 and 0.38 at willingness-to-pay values of €0, €20,000 and €50,000 per QALY gained, respectively.

## Sensitivity analyses

Performing the cost-effectiveness analysis without adjustment for confounders (SA1) gives relatively similar results (Table 2). However, the QALY difference increased from -0.037 to -0.081, which impacted the ICER (€8867 per QALY lost instead of €19481 per QALY lost). Including the outlier (SA2) had a much larger impact on the cost-effectiveness results. Although the QALY difference remained relatively similar (-0.037 to -0.033), the cost saving increased from €719 to €2993. This was due to a patient in the usual care group that received 20 hours of home care per day for a consecutive 24 weeks. Consequently, it notably changed the ICERs (€89376 per QALY lost instead of €19481 per QALY lost) and the distribution of the bootstrapped cost-effect pairs on the CE-plane.

## Discussion

Overall, the SMILE intervention resulted in lower costs over the 12-month follow-up period compared to the usual care group, but not in more QALYs gained. For body weight change the SMILE-intervention was dominant over usual care with the probability of the intervention being cost-effective compared to usual care being 92% at a ceiling ratio of 1,000 euro per additional kilogram body weight loss.

The SMILE intervention was not more effective in improving quality of life. Quality of life is a frequently reported patient-reported outcome in lifestyle interventions in this target population. However, a systematic review showed that most studies did not report a clinically

relevant effect on this outcome [15]. Two studies included in this systematic review described large effects. The lifestyle interventions in these studies consisted of high intensity exercise activities that included social interactions, such as soccer practice. The SMILE intervention included low intensity activities (primarily walking) and social activities, as it was conducted in groups. Although these were important aspects of the intervention, there was less focus on social activities and the intensity of exercise was lower compared to the two studies in the systematic review that showed large effects on quality of life [15].

Further evidence shows that the EQ-5D has adequate item functioning, reliability, validity and responsiveness for patients with mild, moderate and major depression [49, 50]. However, studies are inconclusive about the use of the EQ-5D questionnaire for people with schizophrenia, which is the most common diagnosis within the SMI population [50, 51]. Using the EQ-5D in SMI populations has been shown to be challenging [52]. Research has indicated that available generic quality of life measures may not be appropriate for people with mental health problems [53, 54]. Future studies could consider the use of bolt-on dimensions when working with the EQ-5D by adding additional dimensions relevant to the SMI population to the existing five dimensions of the measure (mobility, self-care, usual activities, pain/discomfort and anxiety/depression). Using bolt-on dimensions makes the questionnaire more responsive for other dimensions, such as psycho-social well-being, vitality, and relationship dimensions that are not included in the EQ-5D, and possibly make a better 'fit' for people with SMI [55, 56]. As found in the in the qualitative assessment of experiences and perceptions of the SMILE intervention, people with SMI indicated that especially feeling safe and belonging to a group was what they liked the most about the intervention [24, 25]. This could possibly be captured by dimensions such the aforementioned bolt-on dimensions (psycho-social wellbeing and relationships). Although these shortcomings of the EQ-5D within the SMI population are recognized, for multiple reasons, we still opted for the use of the EQ-5D. Namely, within economic evaluations estimating the QALY using the EQ-5D is common practice [57], a Dutch tariff is available to estimate utilities from the EQ-5D [33], using a QALY as an outcome measure allows for comparison between different disease areas and societal (Dutch and international) willingness-to-pay thresholds have been based on the QALY [48, 57]. And lastly, this project was funded by ZonMW (Netherlands Organization for Health Research and Development) that requires using the QALY (EQ-5D) in economic evaluations.

Interestingly, in the SMILE intervention group we found lower costs for home care (-€1724 (95%CI -6519; 504) and higher costs for informal care €611 (95%CI 165; 1117) in comparison to the usual care group. In the Netherlands, people with SMI only receive home care after referral by their healthcare provider. Informal care is given by relatives and friend and can be arranged by people with SMI themselves. According to the process evaluation and a qualitative study into experiences and perceptions of people with SMI receiving the SMILE intervention, people with SMI indicated that the intervention had positive impact on their lifestyle [24, 25]. Examples that were mentioned by people with SMI were that they felt better, woke up earlier, organized and cleaned their homes and overall improved their day-to-day routine. This means that patients might have become more independent due to the intervention and/or more empowered to ask for and receive more informal care. Consequently, this may have led to a reduction of the need for homecare. However, we did not expect such a large difference in home care and informal care costs and this would be a topic for further research.

## Strengths and limitations

Our study was performed with a pragmatic design in a real world setting with the same available resources and time as usual practice and, therefore, contributing to the generalizability of

results. In addition, a societal perspective was used in this study. This entails that all costs were collected regardless of who had paid for them. This provides the opportunity to identify shifting of costs between sectors (e.g., healthcare and social affairs and employment). Furthermore, we performed multiple imputation and used a multi-level regression analysis to account for clustering in the data, which is recommended in literature but not commonly performed [42, 58].

There are also some limitations of this study. We found an outlier that strongly influenced the results. One patient in the control group received 20 hours of home care for a consecutive 24 weeks each day. With this outlier included, cost saving increased from €719 to €2993, which consequently changed the ICERs. We asked healthcare professionals what could have been the reason for this high amount of home care, and they considered it to be a faulty entrance of the data. We therefore excluded this patient from the main analysis. Furthermore, although the sample size was smaller than planned (224 participants instead of 260), we found a statistically significant difference in the primary outcome of the trial (body weight loss). A larger sample size would have resulted in more precise estimates of costs and effects (somewhat smaller 95% CIs), but most likely would not change the conclusion.

## Implications for future research, practice and policy makers

Knowledge about the cost-effectiveness of lifestyle interventions for people with SMI is essential for informing policy makers. Various studies of lifestyle intervention for people with SMI have been performed, however, they mainly focus on the effectiveness of the lifestyle intervention. Currently, literature on cost-effectiveness analysis of lifestyle interventions for people with SMI is scarce [13, 21, 22]. Therefore, this study offers an important contribution to the scarce literature available. Furthermore, our results were very much in line with cost-effectiveness analysis of lifestyle interventions in the Netherlands, that is, the lifestyle interventions resulted in non-significant lower costs than usual care and no significant difference in quality of life was found [19, 59].

It is known that people with SMI may need more support in comparison to the general population to overcome barriers associated with their health conditions such as cognitive deficits [60] and lower health literacy [61], which presumably translates into more investment and thus higher costs. It is promising that we found lower costs for the SMILE intervention compared to usual care, making lifestyle interventions targeting people with SMI more of interest. In the Netherlands, where FACT-teams are an important healthcare setting for people with SMI, implementation of the SMILE lifestyle intervention might be opportune if weight loss is considered the most important outcome, especially considering the results of our process evaluation and qualitative study. The intervention was well perceived by clients and mental healthcare professionals who delivered the intervention. They also considered the SMILE intervention feasible [24, 25]. However, it does not seem opportune to implement the SMILE intervention if QALYs are considered more important. Furthermore, one of the key barriers for implementing such lifestyle interventions are that people with SMI often mentioned that their cognitive impairments as a barrier to effective participation in the intervention. This is in line with literature, where cognitive deficits, negative symptom and lower health literacy are recognized as challenges for the implementation of lifestyle interventions in this population. One of the key facilitators was the group-based setting of the intervention, making people with SMI feel safe with their healthcare provider and be part of a group. Further identified barriers and facilitators of the SMILE intervention can be found elsewhere [24, 25]. Although the results of this study might be somewhat generalizable to other countries and settings, more cost-effectiveness studies in other countries and other settings are needed to gain further

insight into the cost-effectiveness of lifestyle interventions for people with SMI while taking into consideration the identified barriers and facilitators of the implementation of such interventions. This is especially needed since ambulatory mental healthcare might be organized differently compared to the Netherlands.

## Conclusion

The SMILE intervention was less costly compared to usual care and cost-effective for body weight change. However, the SMILE intervention does not seem to be cost-effective when QALYs are the main outcome.

## Supporting information

**S1 Checklist. CHEERS 2022 checklist.**
(DOCX)

**S2 Checklist. CONSORT 2010 checklist of information to include when reporting a randomised trial\*.**
(DOC)

**S1 Data.**
(DTA)

## Author Contributions

**Conceptualization:** Berno van Meijel, Maurits W. van Tulder, Marcel Adriaanse.

**Data curation:** Berno van Meijel, Marcel Adriaanse.

**Formal analysis:** Mohamed El Alili.

**Funding acquisition:** Berno van Meijel, Maurits W. van Tulder, Marcel Adriaanse.

**Investigation:** Marcel Adriaanse.

**Methodology:** Mohamed El Alili, Maurits W. van Tulder, Marcel Adriaanse.

**Project administration:** Berno van Meijel, Maurits W. van Tulder, Marcel Adriaanse.

**Resources:** Berno van Meijel, Marcel Adriaanse.

**Software:** Mohamed El Alili, Marcel Adriaanse.

**Supervision:** Berno van Meijel, Maurits W. van Tulder, Marcel Adriaanse.

**Writing – original draft:** Mohamed El Alili, Maurits W. van Tulder, Marcel Adriaanse.

**Writing – review & editing:** Mohamed El Alili, Berno van Meijel, Maurits W. van Tulder, Marcel Adriaanse.

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
