## [Decision Letter · Decision Letter 0]

24 Mar 2024

PONE-D-23-37721Cost-effectiveness of the SMILE intervention compared with usual care for people with severe mental illness: A Randomized Clinical TrialPLOS ONE

Dear Dr. El Alili,

Thank you for submitting your manuscript to PLOS ONE. After careful consideration, we feel that it has merit but does not fully meet PLOS ONE’s publication criteria as it currently stands. Therefore, we invite you to submit a revised version of the manuscript that addresses the points raised during the review process.

We look forward to receiving your revised manuscript.

Kind regards,

Hariom Kumar Solanki, M.D.

Academic Editor

PLOS ONE

Journal Requirements:

We note that you have indicated that there are restrictions to data sharing for this study. For studies involving human research participant data or other sensitive data, we encourage authors to share de-identified or anonymized data. However, when data cannot be publicly shared for ethical reasons, we allow authors to make their data sets available upon request. For information on unacceptable data access restrictions, please see http://journals.plos.org/plosone/s/data-availability#loc-unacceptable-data-access-restrictions.   

Reviewers' comments:

Reviewer's Responses to Questions

**Comments to the Author**

1. Is the manuscript technically sound, and do the data support the conclusions?

Reviewer #1: Yes

Reviewer #2: Yes

Reviewer #3: Partly

Reviewer #4: Yes

2. Has the statistical analysis been performed appropriately and rigorously? 

Reviewer #1: Yes

Reviewer #2: Yes

Reviewer #3: No

Reviewer #4: Yes

3. Have the authors made all data underlying the findings in their manuscript fully available?

Reviewer #1: Yes

Reviewer #2: Yes

Reviewer #3: No

Reviewer #4: No

4. Is the manuscript presented in an intelligible fashion and written in standard English?

Reviewer #1: Yes

Reviewer #2: Yes

Reviewer #3: Yes

Reviewer #4: Yes

5. Review Comments to the Author

Reviewer #1: Interesting analysis of an interesting cluster randomized study. Although the design paper was previously published, I would like to see some brief discussion of the sample size (you still discuss randomization), and also in the conclusions, some discussion as to whether the assumptions on the sample size were realized in the study (by this I mean "Assuming an ICC of 0.01 and a dropout rate of 20%."

Reviewer #2: Reviewer Comment to Authors:

This manuscript presents a well-conducted economic evaluation of a lifestyle intervention for individuals with severe mental illness (SMI) in the Netherlands, filling a crucial research gap. The authors have employed appropriate methods, including a pragmatic cluster-randomized controlled trial design and robust statistical analyses.

Areas Requiring Attention:

1. Quality of Life Measurement:

The lack of improvement in quality-adjusted life years (QALYs) warrants further exploration. A more comprehensive discussion and rationale for the choice of quality of life measure would enhance the manuscript.

2. Unexpected Cost Differences:

The unexpected disparities in home care and informal care costs require further investigation. Additional qualitative or process evaluation data could offer insights into these cost differences.

3. Outlier Analysis:

Further elucidation on the criteria used to identify and exclude outliers, along with supplementary sensitivity analyses, would strengthen the findings.

4. Generalizability and Implementation:

A more comprehensive examination of the generalizability of findings to other healthcare settings or countries, along with a deeper exploration of barriers and facilitators to implementation, would be beneficial.

Reviewer #3: The title of the article is sound but the pattern of writing is not scientifically rigorous. The are statistically faulty results presented especially the Confidence Intervals presented. Equally, the methodology presented is not sound enough to allow for replication by third-party if needed.

Reviewer #4: I think that the authors should rework the discussion section. It needs more reflection and connecting with articles in the space; the findings from this study should be compared with other studies while highlighting the implications.

6. PLOS authors have the option to publish the peer review history of their article (what does this mean?). If published, this will include your full peer review and any attached files.

Reviewer #1: No

Reviewer #2: No

Reviewer #3: **Yes: **Dr Abdulrahman Ahmad

Reviewer #4: No

---

## [Author Response · Author response to Decision Letter 0]

3 May 2024

Please see the included response to reviewers where we've replied to each comment made.

---

## [Decision Letter · Decision Letter 1]

18 Oct 2024

Cost-effectiveness of the SMILE intervention compared with usual care for people with severe mental illness: A Randomized Clinical Trial

PONE-D-23-37721R1

Dear Dr. El Alili,

We’re pleased to inform you that your manuscript has been judged scientifically suitable for publication and will be formally accepted for publication once it meets all outstanding technical requirements.

Kind regards,

Hesty Utami Ramadaniati, Ph.D

Academic Editor

PLOS ONE

Additional Editor Comments (optional):

Reviewers' comments:

Reviewer's Responses to Questions

**Comments to the Author**

1. If the authors have adequately addressed your comments raised in a previous round of review and you feel that this manuscript is now acceptable for publication, you may indicate that here to bypass the “Comments to the Author” section, enter your conflict of interest statement in the “Confidential to Editor” section, and submit your "Accept" recommendation.

Reviewer #1: All comments have been addressed

Reviewer #5: All comments have been addressed

2. Is the manuscript technically sound, and do the data support the conclusions?

Reviewer #1: (No Response)

Reviewer #5: Yes

3. Has the statistical analysis been performed appropriately and rigorously? 

Reviewer #1: (No Response)

Reviewer #5: Yes

4. Have the authors made all data underlying the findings in their manuscript fully available?

Reviewer #1: (No Response)

Reviewer #5: Yes

5. Is the manuscript presented in an intelligible fashion and written in standard English?

Reviewer #1: (No Response)

Reviewer #5: Yes

6. Review Comments to the Author

Reviewer #1: (No Response)

Reviewer #5: The revised manuscript has been much improved according to the reviewers' helpful comments.

This reviewer does not have any further comments.

7. PLOS authors have the option to publish the peer review history of their article (what does this mean?). If published, this will include your full peer review and any attached files.

Reviewer #1: No

Reviewer #5: No

---

## [Editor Report · Acceptance letter]

4 Nov 2024

PONE-D-23-37721R1 

PLOS ONE

Dear Dr. El Alili, 

I'm pleased to inform you that your manuscript has been deemed suitable for publication in PLOS ONE. Congratulations! Your manuscript is now being handed over to our production team.

Kind regards, 

on behalf of

Dr. Hesty Utami Ramadaniati 

Academic Editor

PLOS ONE